# Benchmarking probabilistic machine learning in protein fitness landscape predictions

**Ningning Chen** [1]  **Wenkai Han** [2]  **Sai T Reddy** [1]

## Abstract

Machine learning guided protein engineering, which consists of high-throughput screening and deep sequencing of protein mutagenesis libraries combined with machine learning is a powerful approach for engineering proteins and interrogating their fitness landscapes. Uncertainty quantification enhances the trustworthiness of model predictions by indicating reliability and thus can be used to guide downstream experimental work. Aleatoric uncertainty identifying inherent observational noise in protein properties and epistemic uncertainty revealing gaps in the model's knowledge based on the amount of training data. Although uncertainty quantification has been investigated in the application of protein engineering, systematic benchmarks for probabilistic machine learning model selection and the benefits of different types of uncertainty in protein fitness predictions are lacking. Addressing this gap, our study benchmarks six advanced probabilistic modeling techniques across eleven diverse protein-fitness datasets, employing evaluation metrics on prediction accuracy and uncertainty quality to assess performance for both in-distribution and out-of-distribution scenarios. Our findings offer valuable insights into the application of uncertainty-aware machine learning in high-throughput protein screening experiments. Our study supports more robust, efficient experimental processes and enhances the practical usability of machine learning models in real-word protein fitness related tasks such as therapeutic antibody optimization and viral evolution.

[*]Equal contribution  [1]ETH Zurich, Department of Biosystems Science and Engineering, Basel 4056, Switzerland [2]King Abdullah University of Science and Technology, CBRC. Correspondence to: Sai T Reddy <sai.reddy@bsse.ethz.ch>.

*Accepted at the 1st Machine Learning for Life and Material Sciences Workshop at ICML 2024*. Copyright 2024 by the author(s).

## 1. Introduction

Depicting protein fitness landscapes from amino acid sequences is crucial for both evolutionary biology and protein engineering. Traditional experimental approaches to characterize protein fitness can be laborious and resource intensive. For example, high-throughput methods such as deep mutational scanning (DMS) rely on high-throughput screening and deep sequencing of mutagenesis libraries(Fowler & Fields, 2014). However, despite being relatively high-throughput compared to other assessment methods, these approaches still cannot feasibly scan the entire combinatorial sequence space of target proteins. The integration of machine learning with such screening methods has emerged as a valuable approach to interrogating combinatorial protein sequence space at greater depth by predicting the fitness of sequences not captured by experiments and extrapolating higher-order mutations(Yang et al., 2019). While Integrating machine learning into protein fitness landscape predictions requires bridging computational and laboratory efforts, the reliability of the model and the quality of the data should inform further steps and reduce probability of experimental failures and model overfitting. Identifying the aleatoric and epistemic uncertainty can indicate the noise inherent in the observations and the model confidence based on the knowledge learned from training data(Kendall & Gal, 2017). In the scenario of protein engineering, quantification of uncertainty could also help to guide subsequent validation experiments by balancing the exploration–exploitation trade-off(Vornholt et al., 0). Several studies have attempted to apply uncertainty-aware methods to protein engineering. However, these efforts often involve single probability machine learning models that narrowly address specific goals(Yang et al., 2019; Zeng & Gifford, 2019) or focus on downstream applications without systematically estimating uncertainty quality(Hie et al., 2020).Greenman et al. provided a benchmark for uncertainty quantification in protein engineering based on three datasets and tested it in an active learning setting (Greenman et al., 2023). The study concluded that none of the methods tested preform consistently well and the quality of uncertainty does not necessarily correlate to gains provided by active-learning. Thus a comprehensive framework is still needed to fully evaluate the potential benefits of aleatoric and epistemic uncertainty Ad-

ditionally, methods incorporating Bayesian neural networks have not yet been applied to protein fitness-related tasks. To address these gaps we predict protein fitness by leveraging the following probabilistic approaches: Bayes by Backprop, Gaussian Process, Mean-Variance Estimation, Deep Ensemble, Monte Carlo dropout, and Stochastic Weight Averaging. These methods not only provide a prediction but also an uncertainty score. Furthermore, predictive uncertainty can be decomposed into aleatoric and epistemic types. Aleatoric uncertainty encompasses inherent data noise, which can arise from both biological variability, such as protein stability, and non-biological variability, such as measurement errors. Epistemic uncertainty, on the other hand, results from a lack of sufficient training data.This approach enables us to address data noise and model generalization while guiding subsequent experimental designs to efficiently improve protein functionality with reduced cost and time.

The main contributions of this study are :

- Benchmark different types of probabilistic modeling methods first in synthetic toy dataset and then in real-world protein fitness prediction task.

- Decompose the uncertainty into aleatoric and epistemic and test how they are related to noise in the dataset and model's extrapolation ability.

- Provide insights in how to select probabilstic modeling methods in protein fitness prediciton tasks.

## 2. Problem setup

Let $x$ denote the onehot encoding of an amino acid sequence, and $y$ as the fitness value of the protein from a high-throughput screening experimental assessment, where $y \in \mathbb{R}$ for regression. The parameter $\theta$ indicates the whole parameters used in a machine learning model. Our goal is to learn a predictive model $p(\theta \mid x_{1:n}, y_{1:n})$ based on the training samples $D = \{(x_n, y_n)\}_{n=1}^{N}$. The probabilistic model output will be the mean value $\mu$ and the standard deviation $\sigma$ instead of a single value predicted by point estimation.

## 3. Experimental setup

Figure1 provides an illustrative overview of the pipeline of applying probabilistic modeling in protein fitness landscape predictions. There are six probabilistic modeling algorithms tested plus the multilayer perception as the deterministic baseline. After hyper-parameter searching for each dataset, we evaluate both the mean prediction and uncertainty quality using nice metrics. Additionally, we further decompose the uncertainty score into aleatoric uncertainty and epistemic uncertainty, representing the uncertainty caused by the noise

inherent in the dataset and the lack of data respectively:

$$\text{Var}[y^* \mid \mathbf{x}_{1:n}, \mathbf{y}_{1:n}, \mathbf{x}^*] = \underbrace{\mathbb{E}\left[\text{Var}[y^* \mid \mathbf{x}^*, \theta]\right]}_{\text{Aleatoric uncertainty}}$$
$$+ \underbrace{\text{Var}\left[\mathbb{E}[y^* \mid \mathbf{x}^*, \theta]\right]}_{\text{Epistemic uncertainty}} \quad (1)$$

In this study, we benchmark the probabilistic modeling in both synthetic toy dataset and the real world protein fitness landscape task with datasets obtained via deep mutational scanning.

### 3.1. Synthetic dataset

The toy setting uses a Gaussian Process to generate synthetic data $D = \{(x_n, y_n)\}_{n=1}^{N}$. To match the fact that there exists noises in the deep mutational scanning, we also introduce the heteroskedastic noise, which can be aligned with the aleatoric uncertainty in equation 1. We design two evaluation settings to test the uncertainty quality and uncertainty separation in in-distribution(ID) scenario and out-of-distribution(OOD) scenario. The training dataset is sampled in certain range of $x$, ID test set would also be sampled from that range and the OOD test set would be sampled outside of that region (figure1b bottom panel).

### 3.2. Protein fitness dataset

We perform our computational analysis using ProteinGym(Notin et al., 2024), which curated over 250 standardized DMS assays for different types of proteins and their mutation effects related to protein properties such as binding affinity to a target. To evaluate probabilistic modeling for both ID and OOD, we only retain datasets that contain mutations larger than two. This results in 11 DMS datasets targeting various sequence lengths and functions of proteins, including properties such as binding affinity and fluorescence of GFP (Table2). Single and double mutations can often be recovered in DMS experiments; however, typically higher-order mutations (i.e. triple or more) are not captured by DMS due to exponential diversity of combinatorial sequence space. To align with this situation, we train our models on data with mutations involving fewer than two positions and test the model completion ability within two or fewer mutations (ID test setup). We also test the model's extrapolation ability on the mutations involving more than two positions (OOD test setup). Additionally, the amount of data varies among the DMS datasets. To simulate realistic conditions where data collection is expensive and time-consuming, and to reduce computational costs during training, we restrict the training data to 1000 samples for any dataset exceeding this number. This constraint ensures the training process operates within a low data regime, which can be processed in a reasonable amount of time.

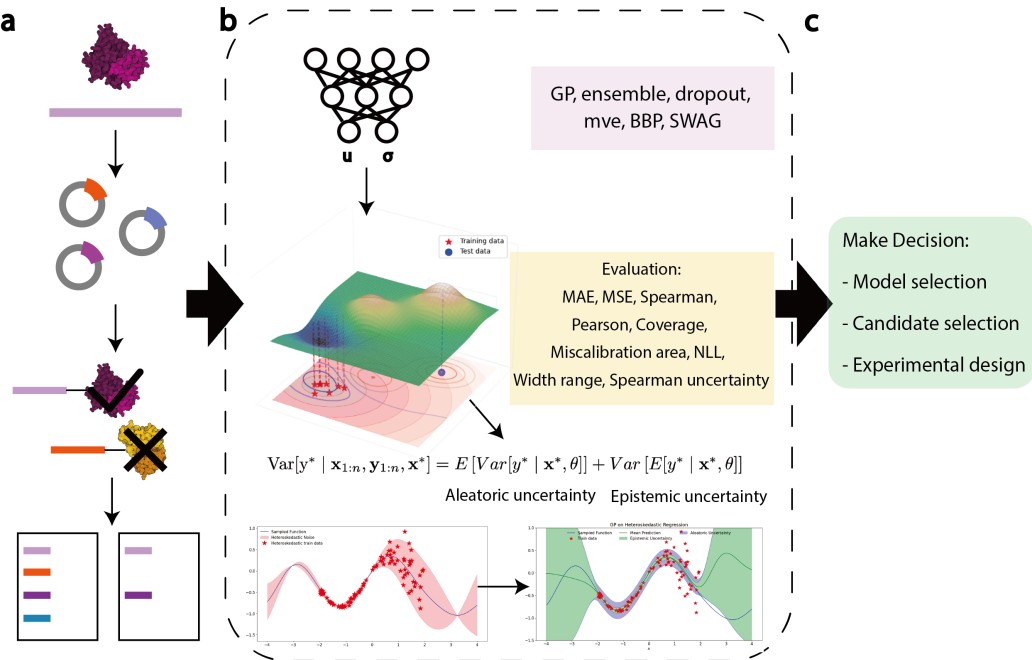

*Figure 1.* **Illustration of the pipeline for leveraging the probabilistic modeling in protein fitness landscape prediction. a**. Schematic of the approach for deep mutational scanning. Basically, it includes generating plasmids that encode the mutated target protein, choosing a proper system to display the protein, selection based on the interested property and comparing the sequencing results between pre-selection and post-selection. **b**. The modeling step after collecting the data. A panel of probabilistic modeling methods are evaluated using multiple metrics in both in-distribution test set and out-of-distribution test set. The uncertainty is decomposed into aleatoric uncertainty and epistemic uncertainty and evaluated separately. **c**. Decision making after the modeling training and evaluation.

## 3.3. Probabilistic modeling methods

Here we provide a high-level overview of the probabilistic modeling methods used in this study, leaving the detailed description to the supplementary material.**Gaussian Process (GP)** is a non-parametric model that specifies a multivariate normal distribution over functions. We use the GP model with the RBF kernel and the likelihood function as Gaussian noise(MacKay et al., 1998). **Bayes by Backprop (BBP)** is a Bayesian neural network model that uses Gaussian reparameterization trick to approximate the posterior distribution of the weights directly through backpropagation(Blundell et al., 2015). Instead of a single value prediction, **Mean-Variance Estimation (MVE)** outputs the parameters of a Gaussian distribution with mean and variance. **Deep Ensemble (Ensemble)** uses an ensemble of neural networks instead of a single one. Here, we also apply the bootstrap method to generate different training sets for each network, enhancing robustness and improving uncertainty estimates(Lakshminarayanan et al., 2017). **MC Dropout(Dropout)** approximates a Bayesian inference by randomly dropping out neurons in multiple forward passes, providing a means to estimate the uncertainty in the model's predictions (Gal & Ghahramani, 2016). **Stochastic Weight Averaging (SWAG)** estimates the Gaussian pos-

teriors over BNN weights by utilizing stochastic weight averaging (Zellers et al., 2018).

## 3.4. Evaluation metrics

We evaluate the models using the following metrics: Mean Absolute Error (MAE), Mean Squared Error (MSE), Pearson correlation coefficient (Pearson), Spearman correlation coefficient (SPCC), Percentage of coverage of 95% confidence interval (Per Co), average width relative to range (AVR), Spearman correlation of uncertainty and error (SPCC unc), Negative Log-Likelihood (NLL), and miscalibration area (Mis area), which are well defined in other UQ studies(Tran et al., 2020; Greenman et al., 2023) and further details can be found in the appendix.

## 4. Results

**Synthetic dataset**: Table 1 presents the performance of the probabilistic modeling methods on the synthetic dataset. In the in-distribution (ID) test setting, the Gaussian Process (GP) model and the Mean-Variance Estimation (MVE) model outperform the other models in terms of Mean Absolute Error (MAE) and Pearson correlation. However, while the GP model's uncertainty quality is generally poor

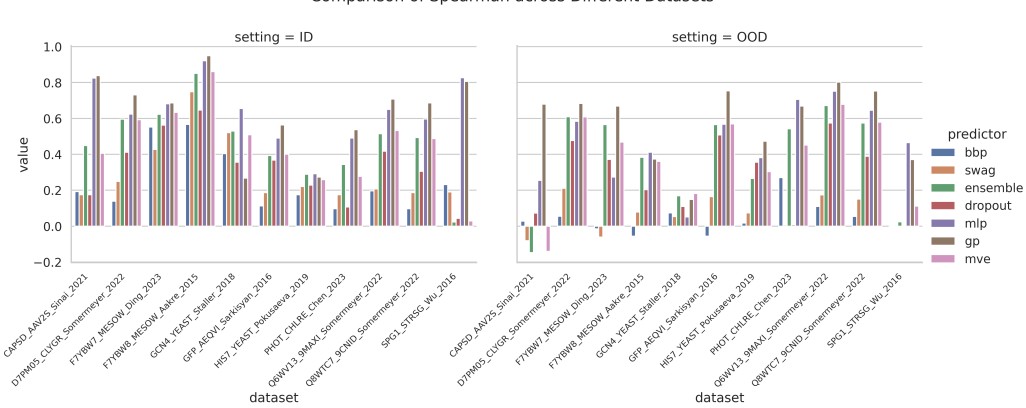

*Figure 2.* Spearman correlation of the probabilistic modeling methods on the protein fitness datasets in both ID and OOD settings.

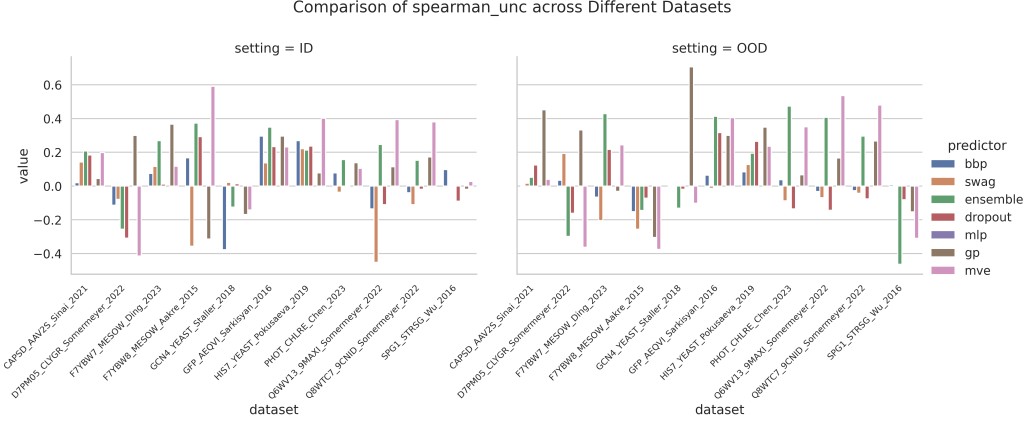

*Figure 3.* Spearman correlation of the probabilistic modeling methods on the protein fitness datasets in both ID and OOD settings.

compared to the others, the MVE model excels in uncertainty evaluation. The ensemble method shows performance slightly inferior to the MVE model in both prediction accuracy and uncertainty evaluation. In the out-of-distribution (OOD) test setting, the ensemble approach emerges as the best method, indicating its robustness in extrapolation scenarios. Although the MVE model provides good prediction performance, it fails to accurately identify uncertainty.

We further decompose the uncertainty into aleatoric and epistemic uncertainty (Figure 4). For both the GP and ensemble models, the epistemic uncertainty is well estimated in both ID and OOD settings. However, while the GP model struggles to capture aleatoric uncertainty, the ensemble model appears to depict aleatoric uncertainty to some extent (Figures 4a and 4b). Nonetheless, the relationship between aleatoric uncertainty and the true noise is not clear (Figure 4c), suggesting that aleatoric uncertainty may only provide a rough estimation of the true noise, capturing some trends but lacking granularity. Regarding epistemic uncer-

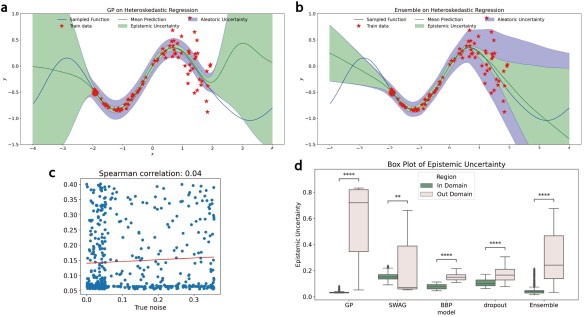

*Figure 4.* The model evaluation on the toy datasets. **a.** The predicted mean and uncertainty for GP model and **b.** for ensemble model. **c.** The scatter plot of aleatoric uncertainty and true noise for ensemble model. **d.** The boxplot of epistemic uncertainty of ID and OOD test sets for GP, swag, BBP, dropout and ensemble models.

*Table 1.* Model evaluation on the synthetic dataset

| Method | MAE | MSE | Pearson | SPCC | Per_Co | AVR | SPCC_unc | NLL | Mis_area |
|---|---|---|---|---|---|---|---|---|---|
| **In Distribution** | | | | | | | | | |
| swag | 0.140 | 0.050 | 0.910 | 0.920 | 0.990 | 0.290 | 0.530 | 0.450 | 0.130 |
| dropout | 0.120 | 0.040 | 0.910 | 0.930 | **1.000** | 0.290 | 0.670 | 0.470 | 0.210 |
| ensemble | 0.120 | 0.040 | 0.910 | 0.930 | 0.980 | 0.200 | 0.620 | 0.840 | 0.100 |
| gp | **0.110** | 0.040 | **0.920** | **0.940** | 0.800 | **0.110** | 0.070 | -0.660 | 0.110 |
| mlp | 0.180 | 0.050 | 0.880 | 0.830 | - | - | - | - | - |
| mve | **0.110** | **0.030** | **0.920** | 0.930 | 0.970 | 0.180 | **0.750** | **0.980** | 0.080 |
| bbp | 0.280 | 0.110 | 0.740 | 0.650 | **1.000** | 0.470 | 0.420 | -0.280 | **0.050** |
| **Out of Distribution** | | | | | | | | | |
| swag | 0.450 | 0.240 | -0.860 | -0.670 | 0.660 | **0.320** | 0.390 | -2.280 | 0.320 |
| dropout | 0.440 | 0.230 | -0.320 | -0.270 | 0.700 | 0.350 | 0.370 | -1.100 | 0.320 |
| ensemble | **0.250** | **0.110** | **0.750** | **0.720** | 0.730 | 0.480 | -0.040 | **-0.900** | **0.090** |
| gp | 0.630 | 0.630 | -0.760 | -0.790 | **0.790** | 0.730 | **0.420** | -1.150 | 0.120 |
| mlp | 0.540 | 0.360 | -0.480 | -0.540 | - | - | - | - | - |
| mve | 0.310 | 0.180 | **0.750** | **0.720** | 0.510 | 0.380 | -0.130 | -448.68 | 0.160 |
| bbp | 0.880 | 0.820 | -0.790 | -0.720 | 0.150 | 0.360 | -0.350 | -5.080 | 0.460 |

tainty, the boxplot in Figure 4d shows that it is generally higher in the OOD setting than in the ID setting. This is expected, as the model is more uncertain in extrapolation scenarios.

**Protein fitness dataset** Tables 3 to 13 and Figures 2 and 3 present the performance of the probabilistic modeling methods on the protein fitness datasets. While no single model consistently outperforms the others across all datasets, some trends are evident. The Gaussian Process (GP) model generally performs the best in both model prediction and uncertainty quality, achieving the top results in 8 out of 11 datasets in both ID and OOD settings.

The point estimation method, Multi-Layer Perceptron (MLP), demonstrates robust performance in the ID setting but sometimes drops significantly in OOD settings. In contrast, models like GP maintain similar performance across both ID and OOD settings (e.g., datasets CAPSD_AAV2S_Sinai_2021 and F7YBW7_MESOW_Ding_2023). This suggests that incorporating uncertainty estimation into modeling can enhance extrapolation ability and mitigate overfitting by attributing error to larger variance. The MVE and ensemble approaches also show relatively good performance in both ID and OOD settings, albeit slightly lower than the GP method. Interestingly, they demonstrate good uncertainty estimation, as indicated by the Spearman correlation between uncertainty and error.

Other methods like Bayes by Backprop (BBP) and Stochastic Weight Averaging-Gaussian (SWAG) perform relatively worse in both prediction and uncertainty estimation. In real-world applications such as protein engineering or drug discovery, the confidence region of the prediction is often more relevant than point estimation. We report the percentage of coverage and the average width of the 95% confidence interval for each model to align with real-world applications. These two metrics should be considered together due to the trade-off between them: larger coverage and narrower width are preferred, but achieving both simultaneously is challeng-

ing. GP tends to have narrower widths but lower coverage, while ensemble and MVE methods have larger coverage but wider widths (Figure 7). Our analysis also shows that different metrics for evaluating uncertainty estimation are not always consistent. For example, while the GP models excel in terms of Negative Log-Likelihood (NLL) and mis-calibration area, their Spearman correlation of uncertainty collapses (Table 5, OOD setting).

We further explore uncertainty types using the CAPSD_AAV2S_Sinai_2021 dataset, which has a large amount of data in both ID and OOD settings and contains a diverse number of mutations. For epistemic uncertainty, we expect higher values in the OOD setting compared to the ID setting, as the model has not seen the data in the OOD test set. The epistemic uncertainty predicted by all models is generally higher in the OOD setting, as expected, with the GP model showing a clear trend of increasing uncertainty with the number of mutations (Figure 5a and 5b). With the better deciption of the epistemic uncertainty, GP also shows highest generalization ability in this case(Table 4). For aleatoric uncertainty, we assume it correlates with the inherent noise in the datasets, caused by factors such as experimental noise, protein stability, and the data collection process. We tested the relationship between aleatoric uncertainty and mutational position and found some variability, but further investigation of positions with higher or lower aleatoric uncertainty is needed to draw definitive conclusions (Figure 6).

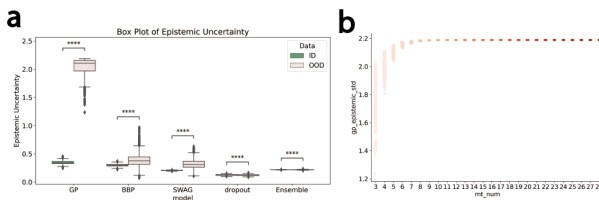

*Figure 5.* **a.** Epistemic uncertainty comparison between ID and OOD **b.** Epistemic uncertainty from GP vs the number of mutations.

## 5. Discussion

In this study, we benchmarked six advanced probabilistic modeling techniques across eleven protein fitness datasets and a synthetic toy dataset. To evaluate the impact of uncertainty quantification on model performance and generalization, we designed both in-distribution (ID) and out-of-distribution (OOD) settings. Our results reveal that ensemble models can accurately predict aleatoric and epistemic uncertainty in both ID and OOD settings while maintaining high model performance.

However, when applied to real-world protein fitness datasets,

the Gaussian Process (GP) model generally outperforms other models in terms of both prediction performance and uncertainty quality. This may be due to the higher dimensionality of the input data in real-world scenarios, making the GP model more suitable for high-dimensional data. This finding underscores the importance of carefully considering and testing data representations during the model selection process. Instead of one-hot encoding, we could explore representations from pre-trained models, such as large protein language models, which have been shown to benefit downstream tasks (Rives et al., 2021).

Our results also indicate that different metrics for evaluating uncertainty estimation are not always consistent, highlighting the complexity of uncertainty estimation and the need for further investigation. Currently, metric selection should align with the specific goals of the modeling task.

For future work, we plan to test the models with varying training data sizes, hypothesizing that the GP model may be more robust in low-data regimes, whereas Bayesian neural networks and dropout techniques might perform better with larger datasets. Additionally, there is a need for a standardized framework to evaluate model performance and uncertainty estimation to accelerate protein engineering and other real-world applications.

It is also worth exploring the application of uncertainty values in next-round experimental design using strategies such as active learning or Bayesian optimization.

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

# A. Appendix

*Table 2.* Summary of the deep mutational scanning data

| DMS_id | seq_len | region mutated | selection type | selection_assay | id_test | ood_test | train |
|---|---|---|---|---|---|---|---|
| CAPSD_AAV2S_Sinai_2021 | 735 | 561-588 | nan | viability for AAV capsid production | 2273 | 30963 | 1000 |
| D7PM05_CLYGR_Somermeyer_2022 | 235 | 2-235 | FACS | Fluorescence | 2263 | 13198 | 1000 |
| F7YBW7_MESOW_Ding_2023 | 93 | 48-82 | nan | growth enrichment | 33 | 7756 | 133 |
| F7YBW8_MESOW_Aakre_2015 | 93 | 59-64 | Growth | fitness | 107 | 8656 | 429 |
| GCN4_YEAST_Staller_2018 | 281 | 101-144 | FACS | Binding | 18 | 2550 | 70 |
| GFP_AEQVI_Sarkisyan_2016 | 238 | 3-237 | FACS | Fluorescence | 2772 | 37853 | 1000 |
| HIS7_YEAST_Pokusaeva_2019 | 220 | 6-211 | Growth | Growth | 329 | 494494 | 1000 |
| PHOT_CHLRE_Chen_2023 | 118 | 1-118 | FACS | Fluorescence | 460 | 165231 | 1000 |
| Q6WV13_9MAXI_Somermeyer_2022 | 222 | 2-222 | FACS | Fluorescence | 3427 | 14268 | 1000 |
| Q8WTC7_9CNID_Somermeyer_2022 | 238 | 2-238 | FACS | Fluorescence | 2492 | 21049 | 1000 |
| SPG1_STRSG_Wu_2016 | 448 | 265-280 | binding | Binding (IgG) | 433 | 147193 | 1000 |

*Table 3.* Performance Metrics for Different Predictors for D7PM05_CLYGR_Somermeyer_2022

| Method | MAE | MSE | Pearson | SPCC | Per_Co | AVR | SPCC_uncLL | | Mis_area |
|---|---|---|---|---|---|---|---|---|---|
| **In Distribution** | | | | | | | | | |
| bbp | 1.195 | 2.398 | 0.181 | 0.139 | **1.000** | 0.601 | -0.114 | -2.054 | 0.177 |
| ensemble | 1.308 | 2.208 | 0.651 | 0.596 | **1.000** | 0.383 | -0.255 | -1.828 | 0.103 |
| gp | **0.763** | **1.022** | **0.757** | **0.732** | 0.834 | **0.153** | **0.300** | **-1.617** | **0.094** |
| mlp | 0.765 | 1.159 | 0.718 | 0.624 | - | - | - | - | - |
| mve | 1.210 | 2.194 | 0.666 | 0.593 | **1.000** | 0.359 | -0.414 | -1.816 | 0.094 |
| dropout | 1.300 | 2.188 | 0.465 | 0.412 | **1.000** | 0.460 | -0.308 | -1.888 | 0.114 |
| swag | 2.314 | 6.504 | 0.275 | 0.249 | 0.996 | 0.433 | -0.079 | -2.470 | 0.211 |
| **Out of Distribution** | | | | | | | | | |
| bbp | 1.990 | 4.864 | 0.058 | 0.056 | **1.000** | 0.553 | 0.034 | -2.225 | 0.096 |
| ensemble | 1.883 | 4.242 | 0.591 | 0.609 | **1.000** | 0.382 | -0.298 | -2.199 | 0.191 |
| gp | 1.480 | 3.294 | **0.691** | **0.684** | 0.725 | **0.227** | **0.332** | -2.425 | 0.217 |
| mlp | 1.647 | 4.479 | 0.575 | 0.585 | - | - | - | - | - |
| mve | 1.949 | 4.813 | 0.590 | 0.608 | 0.999 | 0.357 | -0.362 | -2.360 | 0.197 |
| dropout | 1.848 | 4.073 | 0.483 | 0.477 | **1.000** | 0.455 | -0.162 | -2.125 | 0.140 |
| swag | **1.404** | **2.784** | 0.231 | 0.211 | 0.997 | 0.423 | 0.194 | **-1.939** | **0.039** |

*Table 4.* Performance Metrics for Different Predictors for CAPSD_AAV2S_Sinai_2021

| Method | MAE | MSE | Pearson | SPCC | Per_Co | AVR | SPCC_uncLL | | Mis_area |
|---|---|---|---|---|---|---|---|---|---|
| **In Distribution** | | | | | | | | | |
| bbp | 2.834 | 10.582 | 0.190 | 0.193 | 0.987 | 0.056 | 0.019 | -2.604 | 0.061 |
| swag | 2.845 | 10.604 | 0.169 | 0.175 | 0.976 | 0.049 | 0.142 | -2.603 | 0.096 |
| ensemble | 2.908 | 11.043 | 0.439 | 0.449 | **0.992** | 0.056 | **0.207** | -2.621 | 0.069 |
| dropout | 2.872 | 10.765 | 0.167 | 0.175 | 0.978 | 0.049 | 0.183 | -2.610 | 0.099 |
| mlp | **1.478** | 3.518 | 0.821 | 0.825 | - | - | - | - | - |
| gp | 1.479 | **3.453** | **0.828** | **0.838** | 0.910 | **0.025** | 0.045 | **-2.084** | **0.056** |
| mve | 2.874 | 10.776 | 0.391 | 0.406 | 0.977 | 0.049 | 0.198 | -2.613 | 0.101 |
| **Out of Distribution** | | | | | | | | | |
| bbp | 2.652 | 8.860 | 0.021 | 0.028 | 0.988 | 0.052 | 0.001 | -2.524 | 0.080 |
| swag | 2.687 | 9.107 | -0.074 | -0.081 | 0.994 | 0.055 | 0.016 | -2.542 | 0.077 |
| ensemble | 2.612 | 8.671 | -0.219 | -0.148 | **0.997** | 0.055 | 0.052 | -2.520 | 0.070 |
| dropout | 2.633 | 8.732 | 0.091 | 0.072 | 0.987 | 0.048 | 0.124 | -2.501 | 0.091 |
| mlp | 4.347 | 30.079 | 0.362 | 0.255 | - | - | - | - | - |
| gp | **2.115** | **7.252** | **0.626** | **0.679** | 0.866 | **0.033** | **0.451** | **-2.449** | **0.069** |
| mve | 2.635 | 8.748 | -0.207 | -0.141 | 0.987 | 0.048 | 0.040 | -2.504 | 0.091 |

*Table 5.* Performance Metrics for Different Predictors for F7YBW7_MESOW_Ding_2023

| Method | MAE | MSE | Pearson | SPCC | Per_Co | AVR | SPCC_uncLL | | Mis_area |
|---|---|---|---|---|---|---|---|---|---|
| **In Distribution** | | | | | | | | | |
| bbp | 0.222 | 0.088 | 0.554 | 0.552 | **1.000** | 1.329 | 0.075 | -0.305 | 0.147 |
| dropout | 0.227 | 0.092 | 0.567 | 0.562 | 0.912 | 0.940 | 0.012 | -0.207 | 0.069 |
| mve | 0.220 | 0.088 | **0.671** | 0.634 | 0.971 | 0.961 | 0.117 | -0.193 | 0.064 |
| ensemble | 0.227 | 0.093 | 0.657 | 0.623 | 0.912 | 0.927 | 0.268 | -0.219 | 0.063 |
| gp | **0.167** | **0.054** | 0.653 | **0.687** | 0.941 | **0.573** | **0.367** | **0.047** | **0.029** |
| swag | 0.231 | 0.096 | 0.332 | 0.428 | 0.882 | 0.916 | 0.117 | -0.231 | 0.087 |
| mlp | 0.179 | 0.060 | 0.657 | 0.681 | - | - | - | - | - |
| **Out of Distribution** | | | | | | | | | |
| bbp | 0.582 | 0.403 | 0.020 | -0.015 | 0.909 | 1.372 | -0.065 | -1.161 | 0.243 |
| dropout | 0.591 | 0.413 | 0.383 | 0.371 | 0.389 | **0.966** | 0.217 | -1.886 | 0.338 |
| mve | 0.557 | 0.366 | 0.452 | 0.468 | 0.508 | 0.995 | 0.244 | -1.563 | 0.326 |
| ensemble | 0.575 | 0.389 | 0.522 | 0.565 | 0.405 | 0.966 | **0.429** | -1.753 | 0.336 |
| gp | **0.250** | **0.083** | **0.703** | **0.669** | **0.979** | 0.988 | -0.031 | **-0.211** | **0.044** |
| swag | 0.591 | 0.414 | -0.064 | -0.060 | 0.516 | 1.030 | -0.205 | -1.831 | 0.320 |
| mlp | 0.535 | 0.382 | 0.370 | 0.273 | - | - | - | - | - |

*Table 6.* Performance Metrics for Different Predictors for F7YBW8_MESOW_Aakre_2015

| Method | MAE | MSE | Pearson | SPCC | Per_Co | AVR | SPCC_uncLL | | Mis_area |
|---|---|---|---|---|---|---|---|---|---|
| **In Distribution** | | | | | | | | | |
| bbp | 0.335 | 0.141 | 0.498 | 0.567 | **1.000** | 1.315 | 0.167 | -0.500 | 0.089 |
| mve | 0.323 | 0.127 | 0.853 | 0.861 | **1.000** | **0.974** | **0.591** | -0.381 | 0.112 |
| gp | **0.094** | **0.016** | **0.949** | **0.950** | **1.000** | 1.430 | -0.314 | **-0.324** | 0.360 |
| ensemble | 0.324 | 0.133 | 0.847 | 0.851 | **1.000** | 1.027 | 0.373 | -0.412 | 0.086 |
| dropout | 0.343 | 0.143 | 0.599 | 0.646 | **1.000** | 1.021 | 0.293 | -0.446 | 0.114 |
| swag | 0.331 | 0.151 | 0.720 | 0.749 | **1.000** | 1.322 | -0.356 | -0.531 | **0.072** |
| mlp | 0.118 | 0.024 | 0.931 | 0.922 | - | - | - | - | - |
| **Out of Distribution** | | | | | | | | | |
| bbp | 0.386 | 0.153 | -0.061 | -0.056 | **1.000** | 1.369 | -0.151 | -0.543 | **0.190** |
| mve | 0.387 | 0.151 | 0.476 | 0.361 | **1.000** | **0.975** | -0.375 | -0.479 | 0.268 |
| gp | **0.081** | **0.014** | **0.579** | 0.374 | **1.000** | 1.824 | -0.305 | -0.552 | 0.402 |
| ensemble | 0.342 | 0.118 | 0.487 | 0.383 | **1.000** | 1.028 | -0.144 | **-0.363** | 0.236 |
| dropout | 0.403 | 0.164 | 0.250 | 0.203 | **1.000** | 1.022 | **-0.072** | -0.517 | 0.273 |
| swag | 0.333 | 0.112 | 0.067 | 0.079 | **1.000** | 1.329 | -0.256 | -0.448 | 0.204 |
| mlp | 0.361 | 0.167 | 0.526 | **0.412** | - | - | - | - | - |

*Table 7.* Performance Metrics for Different Predictors for GCN4_YEAST_Staller_2018

| Method | MAE | MSE | Pearson | SPCC | Per_Co | AVR | SPCC_uncLL | | Mis_area |
|---|---|---|---|---|---|---|---|---|---|
| **In Distribution** | | | | | | | | | |
| bbp | 0.227 | 0.170 | 0.210 | 0.404 | 0.944 | 0.916 | -0.377 | -0.738 | 0.290 |
| mve | 0.242 | 0.147 | 0.347 | 0.509 | 0.944 | 0.378 | -0.141 | -0.552 | 0.118 |
| gp | 0.296 | 0.168 | 0.117 | 0.267 | 0.889 | **0.346** | -0.168 | -1.022 | **0.105** |
| dropout | 0.253 | 0.151 | 0.051 | 0.356 | 0.944 | 0.796 | 0.013 | -0.611 | 0.217 |
| ensemble | 0.246 | 0.147 | 0.314 | 0.529 | 0.944 | 0.403 | -0.125 | **-0.511** | 0.113 |
| swag | 0.570 | 0.364 | 0.263 | 0.521 | **1.000** | 1.078 | **0.022** | -0.996 | 0.131 |
| mlp | **0.211** | **0.122** | **0.465** | **0.655** | - | - | - | - | - |
| **Out of Distribution** | | | | | | | | | |
| bbp | 0.243 | 0.103 | 0.043 | 0.073 | 0.995 | 0.869 | -0.002 | -0.627 | 0.229 |
| mve | 0.204 | 0.077 | **0.101** | **0.182** | 0.987 | **0.403** | -0.102 | **-0.154** | **0.065** |
| gp | 0.718 | 0.769 | 0.089 | 0.149 | 0.812 | 0.700 | **0.707** | -1.384 | 0.174 |
| dropout | 0.204 | 0.079 | 0.084 | 0.109 | 0.994 | 0.794 | -0.018 | -0.529 | 0.252 |
| ensemble | **0.202** | **0.076** | 0.097 | 0.169 | 0.990 | 0.428 | -0.131 | -0.162 | 0.090 |
| swag | 0.447 | 0.271 | 0.048 | 0.053 | **0.998** | 1.022 | 0.001 | -0.895 | 0.097 |
| mlp | 0.268 | 0.127 | 0.026 | 0.052 | - | - | - | - | - |

*Table 8.* Performance Metrics for Different Predictors for GFP_AEQVI_Sarkisyan_2016

| Method | MAE | MSE | Pearson | SPCC | Per_Co | AVR | SPCC_uncLL | | Mis_area |
|---|---|---|---|---|---|---|---|---|---|
| **In Distribution** | | | | | | | | | |
| bbp | 17.182 | 295.678 | 0.166 | 0.113 | **1.000** | - | 0.296 | -19.725 | 0.474 |
| mve | 0.677 | 0.565 | 0.636 | 0.401 | 0.952 | 0.412 | 0.231 | -1.130 | 0.130 |
| ensemble | 0.356 | 0.480 | 0.607 | 0.393 | 0.891 | 0.392 | **0.349** | -1.045 | 0.253 |
| gp | **0.274** | **0.194** | **0.772** | **0.564** | 0.922 | **0.198** | 0.296 | **-0.486** | **0.089** |
| dropout | 0.405 | 0.449 | 0.577 | 0.368 | 0.893 | 0.403 | 0.234 | -1.022 | 0.192 |
| swag | 0.570 | 0.500 | 0.242 | 0.186 | 0.999 | 0.530 | 0.137 | -1.156 | 0.122 |
| mlp | 0.312 | 0.231 | 0.726 | 0.491 | - | - | - | - | - |
| **Out of Distribution** | | | | | | | | | |
| bbp | 15.455 | 241.441 | -0.063 | -0.056 | **1.000** | - | 0.064 | -18.872 | 0.474 |
| mve | 1.022 | 1.322 | 0.585 | 0.570 | 0.751 | 0.414 | 0.405 | -1.780 | 0.224 |
| ensemble | 1.153 | 2.216 | 0.578 | 0.566 | 0.474 | 0.393 | **0.413** | -2.693 | 0.158 |
| gp | **0.550** | **0.487** | **0.758** | **0.753** | 0.922 | **0.341** | 0.300 | **-1.054** | **0.024** |
| dropout | 1.109 | 1.999 | 0.532 | 0.508 | 0.484 | 0.403 | 0.317 | -2.424 | 0.145 |
| swag | 1.089 | 1.783 | 0.177 | 0.165 | 0.937 | 0.564 | -0.013 | -1.789 | 0.099 |
| mlp | 0.859 | 1.368 | 0.589 | 0.568 | - | - | - | - | - |

*Table 9.* Performance Metrics for Different Predictors for PHOT_CHLRE_Chen_2023

| Method | MAE | MSE | Pearson | SPCC | Per_Co | AVR | SPCC_uncLL | | Mis_area |
|---|---|---|---|---|---|---|---|---|---|
| **In Distribution** | | | | | | | | | |
| bbp | 119.690 | 14325.819 | 0.045 | 0.097 | 0.000 | 9.520 | 0.078 | -134.379 | 0.474 |
| mve | 0.302 | 0.133 | 0.214 | 0.277 | 0.943 | 0.464 | 0.105 | -0.408 | 0.046 |
| gp | **0.206** | **0.083** | **0.581** | **0.537** | 0.913 | **0.331** | 0.138 | **-0.175** | 0.050 |
| ensemble | 0.271 | 0.125 | 0.197 | 0.344 | 0.928 | 0.463 | **0.156** | -0.377 | **0.040** |
| swag | 0.404 | 0.199 | 0.059 | 0.175 | **1.000** | 0.647 | -0.036 | -0.624 | 0.091 |
| dropout | 0.291 | 0.129 | 0.102 | 0.107 | 0.952 | 0.524 | 0.002 | -0.409 | 0.050 |
| mlp | 0.249 | 0.103 | 0.493 | 0.490 | - | - | - | - | - |
| **Out of Distribution** | | | | | | | | | |
| bbp | 112.834 | 12737.275 | 0.280 | 0.271 | 0.000 | 7.504 | 0.037 | -233.218 | 0.474 |
| mve | 0.377 | 0.201 | 0.467 | 0.451 | 0.878 | **0.452** | 0.351 | -0.677 | 0.118 |
| gp | 0.352 | 0.171 | 0.651 | 0.669 | 0.995 | 0.568 | 0.067 | -0.530 | **0.044** |
| ensemble | 0.432 | 0.252 | 0.549 | 0.542 | 0.823 | 0.455 | **0.474** | -0.873 | 0.173 |
| swag | **0.297** | **0.131** | 0.005 | 0.006 | **0.999** | 0.613 | -0.088 | **-0.471** | 0.068 |
| dropout | 0.386 | 0.210 | 0.006 | 0.007 | 0.950 | 0.522 | -0.136 | -0.658 | 0.081 |
| mlp | 0.367 | 0.175 | **0.717** | **0.706** | - | - | - | - | - |

*Table 10.* Performance Metrics for Different Predictors for HIS7_YEAST_Pokusaeva_2019

| Method | MAE | MSE | Pearson | SPCC | Per_Co | AVR | SPCC_uncLL | | Mis_area |
|---|---|---|---|---|---|---|---|---|---|
| **In Distribution** | | | | | | | | | |
| bbp | 0.164 | 0.062 | 0.289 | 0.175 | **0.945** | 0.572 | 0.269 | -0.045 | 0.164 |
| mve | 0.122 | 0.055 | 0.680 | 0.259 | 0.942 | 0.460 | **0.401** | 0.093 | 0.222 |
| ensemble | 0.181 | 0.063 | 0.551 | 0.289 | **0.945** | 0.567 | 0.213 | -0.043 | 0.133 |
| gp | 0.116 | 0.032 | 0.705 | 0.274 | 0.894 | **0.284** | 0.078 | **0.247** | **0.079** |
| dropout | 0.140 | 0.059 | 0.515 | 0.229 | 0.942 | 0.514 | 0.236 | 0.007 | 0.194 |
| swag | 0.173 | 0.063 | 0.271 | 0.221 | **0.945** | 0.657 | 0.221 | -0.100 | 0.175 |
| mlp | **0.106** | **0.028** | **0.741** | **0.292** | - | - | - | - | - |
| **Out of Distribution** | | | | | | | | | |
| bbp | 0.385 | 0.301 | 0.025 | 0.018 | 0.657 | 0.549 | 0.084 | -1.523 | 0.104 |
| mve | 0.385 | 0.312 | 0.415 | 0.304 | 0.638 | **0.451** | 0.235 | -2.229 | 0.117 |
| ensemble | 0.375 | 0.262 | 0.314 | 0.266 | 0.666 | 0.562 | 0.194 | -1.210 | 0.076 |
| gp | **0.320** | **0.153** | **0.519** | **0.473** | **0.944** | 0.718 | **0.349** | **-0.469** | **0.030** |
| dropout | 0.380 | 0.298 | 0.414 | 0.356 | 0.652 | 0.511 | 0.264 | -1.669 | 0.107 |
| swag | 0.381 | 0.241 | 0.080 | 0.074 | 0.679 | 0.636 | 0.128 | -0.914 | 0.080 |
| mlp | 0.483 | 0.371 | 0.511 | 0.381 | - | - | - | - | - |

*Table 11.* Performance Metrics for Different Predictors for SPG1_STRSG_Wu_2016

| Method | MAE | MSE | Pearson | SPCC | Per_Co | AVR | SPCC_uncLL | | Mis_area |
|---|---|---|---|---|---|---|---|---|---|
| **In Distribution** | | | | | | | | | |
| bbp | 1.371 | 2.029 | 0.144 | 0.231 | **1.000** | 0.383 | **0.098** | -1.969 | 0.177 |
| gp | 0.205 | 0.206 | 0.785 | 0.807 | 0.977 | 0.122 | -0.019 | **-0.827** | 0.331 |
| mve | 0.358 | 0.487 | -0.122 | 0.029 | 0.938 | 0.107 | 0.027 | -1.058 | 0.206 |
| ensemble | 0.394 | 0.473 | -0.053 | 0.023 | 0.935 | **0.099** | -0.004 | -1.055 | **0.154** |
| swag | 0.412 | 0.470 | 0.251 | 0.191 | 0.945 | 0.109 | -0.000 | -1.042 | 0.160 |
| mlp | **0.190** | **0.137** | **0.846** | **0.827** | - | - | - | - | - |
| dropout | 0.408 | 0.471 | 0.030 | 0.044 | 0.940 | 0.103 | -0.090 | -1.046 | 0.159 |
| **Out of Distribution** | | | | | | | | | |
| bbp | 1.201 | 1.815 | 0.002 | -0.004 | **0.997** | 0.351 | **0.009** | -1.894 | **0.088** |
| gp | **0.082** | **0.141** | 0.370 | 0.371 | 0.987 | 0.132 | -0.153 | -0.832 | 0.436 |
| mve | 0.232 | 0.165 | 0.032 | 0.112 | 0.985 | 0.107 | -0.309 | -0.703 | 0.283 |
| ensemble | 0.302 | 0.189 | -0.003 | 0.024 | 0.984 | **0.099** | -0.464 | **-0.689** | 0.237 |
| swag | 0.319 | 0.197 | -0.006 | -0.005 | 0.986 | 0.109 | -0.006 | -0.751 | 0.235 |
| mlp | 0.146 | 0.141 | **0.435** | **0.466** | - | - | - | - | - |
| dropout | 0.329 | 0.202 | 0.000 | -0.001 | 0.985 | 0.103 | -0.081 | -0.725 | 0.240 |

*Table 12.* Performance Metrics for Different Predictors for Q6WV13_9MAXI_Somermeyer_2022

| Method | MAE | MSE | Pearson | SPCC | Per_Co | AVR | SPCC_uncLL | | Mis_area |
|---|---|---|---|---|---|---|---|---|---|
| **In Distribution** | | | | | | | | | |
| bbp | 1.378 | 2.037 | 0.256 | 0.196 | **1.000** | 0.712 | -0.135 | -2.006 | 0.182 |
| ensemble | 0.806 | 0.950 | 0.738 | 0.515 | 0.904 | 0.278 | 0.246 | -1.391 | **0.106** |
| gp | **0.306** | **0.241** | **0.874** | **0.708** | 0.760 | **0.055** | 0.115 | -2.620 | 0.120 |
| mve | 0.669 | 0.897 | 0.759 | 0.532 | 0.902 | 0.299 | **0.394** | **-1.374** | 0.126 |
| dropout | 0.715 | 0.924 | 0.576 | 0.417 | 0.983 | 0.369 | -0.111 | -1.460 | 0.145 |
| swag | 0.641 | 0.985 | 0.223 | 0.207 | **1.000** | 0.595 | -0.452 | -1.778 | 0.282 |
| mlp | 0.351 | 0.286 | 0.855 | 0.651 | - | - | - | - | - |
| **Out of Distribution** | | | | | | | | | |
| bbp | 1.133 | 1.736 | 0.127 | 0.110 | **1.000** | 0.633 | -0.032 | -1.901 | 0.127 |
| ensemble | 1.020 | 1.615 | 0.731 | 0.671 | 0.772 | 0.279 | 0.407 | -1.729 | 0.079 |
| gp | **0.535** | **0.649** | **0.830** | **0.802** | 0.648 | **0.072** | 0.166 | -4.659 | 0.195 |
| mve | 0.971 | 1.783 | 0.746 | 0.678 | 0.766 | 0.299 | **0.536** | -1.764 | **0.073** |
| dropout | 0.997 | 1.761 | 0.634 | 0.574 | 0.968 | 0.368 | -0.143 | **-1.706** | 0.076 |
| swag | 1.051 | 1.842 | 0.172 | 0.174 | 0.999 | 0.509 | -0.069 | -1.797 | 0.104 |
| mlp | 0.535 | 0.680 | 0.823 | 0.752 | - | - | - | - | - |

*Table 13.* Performance Metrics for Different Predictors for Q8WTC7_9CNID_Somermeyer_2022

| Method | MAE | MSE | Pearson | SPCC | Per_Co | AVR | SPCC_uncLL | | Mis_area |
|---|---|---|---|---|---|---|---|---|---|
| **In Distribution** | | | | | | | | | |
| bbp | 0.851 | 0.922 | 0.137 | 0.097 | **1.000** | 0.565 | -0.040 | -1.702 | 0.189 |
| mve | 0.573 | 0.711 | 0.670 | 0.488 | 0.907 | 0.276 | **0.380** | -1.257 | 0.138 |
| gp | **0.300** | **0.239** | **0.844** | **0.686** | 0.880 | **0.095** | 0.173 | **-0.749** | **0.049** |
| ensemble | 0.871 | 0.920 | 0.657 | 0.494 | 0.988 | 0.297 | 0.152 | -1.380 | 0.128 |
| swag | 1.430 | 2.211 | 0.213 | 0.186 | **1.000** | 0.475 | -0.110 | -1.823 | 0.176 |
| dropout | 0.648 | 0.733 | 0.401 | 0.305 | 0.993 | 0.362 | -0.019 | -1.373 | 0.151 |
| mlp | 0.357 | 0.322 | 0.773 | 0.597 | - | - | - | - | - |
| **Out of Distribution** | | | | | | | | | |
| bbp | 0.822 | 1.459 | 0.058 | 0.054 | 1.000 | 0.638 | -0.027 | -1.851 | 0.229 |
| mve | 0.742 | 1.189 | 0.679 | 0.579 | 0.820 | 0.276 | **0.480** | -1.522 | 0.108 |
| gp | **0.429** | **0.434** | **0.813** | **0.753** | 0.868 | **0.129** | 0.267 | **-1.055** | **0.040** |
| ensemble | 0.913 | 1.098 | 0.634 | 0.575 | 0.980 | 0.297 | 0.296 | -1.465 | 0.088 |
| swag | 1.296 | 1.923 | 0.162 | 0.151 | **1.000** | 0.477 | -0.043 | -1.775 | 0.118 |
| dropout | 0.783 | 1.133 | 0.466 | 0.389 | 0.988 | 0.361 | -0.076 | -1.504 | 0.103 |
| mlp | 0.599 | 0.769 | 0.746 | 0.645 | - | - | - | - | - |

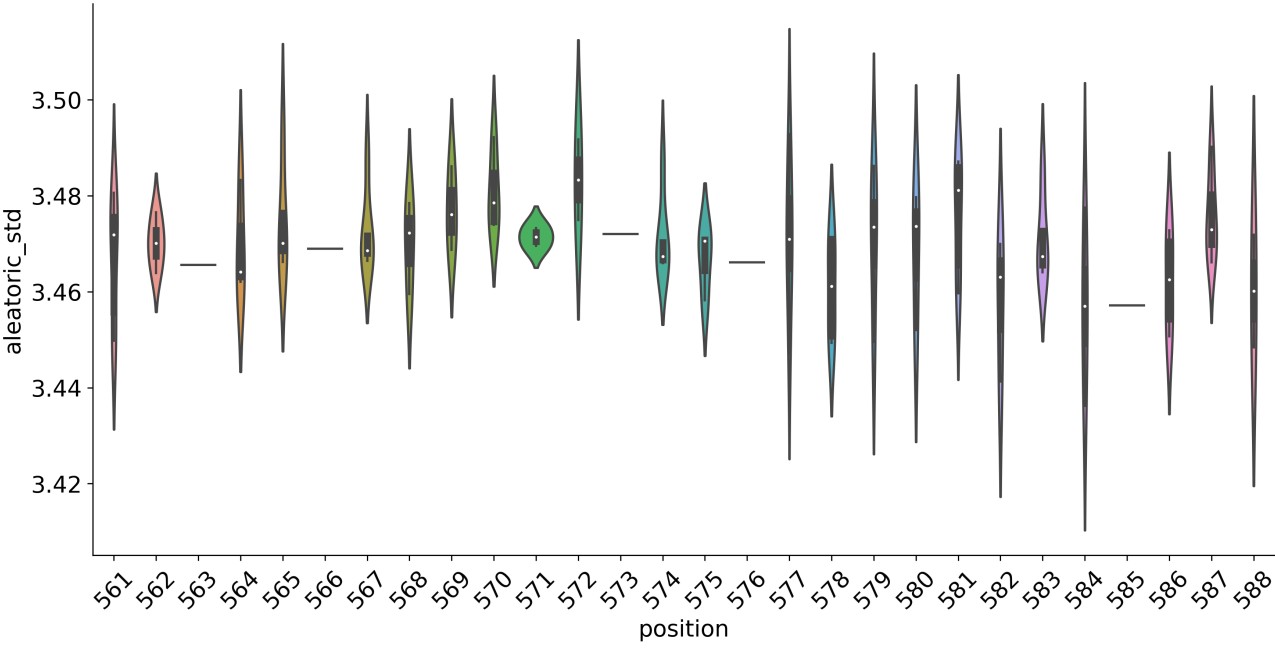

*Figure 6.* Aleatoric Uncertainty vs. Position in the Protein Sequence

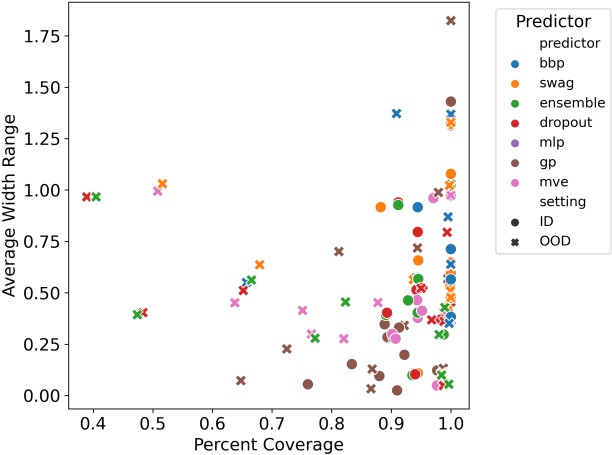

*Figure 7.* Scatter plot of the coverage and width of the 95% confidence interval.

**A.1. Evaluation metrics**

In our study, we use a comprehensive set of metrics to evaluate the performance of our models. These metrics assess both the accuracy of the predictions and the quality of the uncertainty estimates. The metrics are as follows:

**Mean Absolute Error (MAE)**

MAE is the average of the absolute differences between the predicted and actual values. It is defined as:

$$\text{MAE} = \frac{1}{n} \sum_{i=1}^{n} |y_i - \hat{y}_i| \tag{2}$$

where $y_i$ is the actual value and $\hat{y}_i$ is the predicted value.

**Mean Squared Error (MSE)**

MSE is the average of the squared differences between the predicted and actual values. It is defined as:

$$\text{MSE} = \frac{1}{n} \sum_{i=1}^{n} (y_i - \hat{y}_i)^2 \tag{3}$$

**Pearson Correlation Coefficient (Pearson)**

The Pearson correlation coefficient measures the linear correlation between the predicted and actual values. It is defined as:

$$\text{Pearson} = \frac{\sum_{i=1}^{n}(y_i - \bar{y})(\hat{y}_i - \bar{\hat{y}})}{\sqrt{\sum_{i=1}^{n}(y_i - \bar{y})^2 \sum_{i=1}^{n}(\hat{y}_i - \bar{\hat{y}})^2}} \tag{4}$$

where $\bar{y}$ and $\bar{\hat{y}}$ are the means of the actual and predicted values, respectively.

**Spearman Correlation Coefficient (SPCC)**

The Spearman correlation coefficient measures the rank correlation between the predicted and actual values. It is defined as:

$$\text{SPCC} = \frac{\sum_{i=1}^{n}(R(y_i) - \bar{R}(y))(R(\hat{y}_i) - \bar{R}(\hat{y}))}{\sqrt{\sum_{i=1}^{n}(R(y_i) - \bar{R}(y))^2 \sum_{i=1}^{n}(R(\hat{y}_i) - \bar{R}(\hat{y}))^2}} \tag{5}$$

where $R(y_i)$ and $R(\hat{y}_i)$ are the ranks of the actual and predicted values.

**Percentage of Coverage of 95% Confidence Interval (Per_Co)**

This metric measures the percentage of true values that fall within the predicted 95% confidence interval. It is defined as:

$$\text{Per\_Co} = \frac{1}{n} \sum_{i=1}^{n} \mathbf{1}(y_i \in [\hat{y}_i - 1.96\sigma_i, \hat{y}_i + 1.96\sigma_i]) \tag{6}$$

where $\mathbf{1}$ is the indicator function, and $\sigma_i$ is the predicted standard deviation.

**Average Width Relative to Range (AVR)**

AVR measures the average width of the predicted confidence intervals relative to the range of the actual values. It is defined as:

$$\text{AVR} = \frac{1}{n} \sum_{i=1}^{n} \frac{2 \times 1.96\sigma_i}{\max(y) - \min(y)} \tag{7}$$

**Spearman Correlation of Uncertainty and Error (SPCC unc)**

This metric measures the rank correlation between the predicted uncertainties and the absolute errors. It is defined as:

$$\text{SPCC unc} = \text{SPCC}(\sigma_i, |y_i - \hat{y}_i|) \tag{8}$$

**Negative Log-Likelihood (NLL)**

NLL measures the fit of the predicted distribution to the observed data. It is defined as:

$$\text{NLL} = \frac{1}{n} \sum_{i=1}^{n} \left( \frac{1}{2} \log(2\pi\sigma_i^2) + \frac{(y_i - \hat{y}_i)^2}{2\sigma_i^2} \right) \tag{9}$$

**Miscalibration Area (Mis_area)**

The miscalibration area quantifies the discrepancy between predicted and observed confidence intervals. Intuitively, calibration refers that the true value falls within the predicted confidence interval with the same frequency as the predicted confidence level. The miscalibration area identifies the difference between the calibration curve and the ideal calibration curve(Kuleshov et al., 2018).