# OpenReview forum: "Benchmarking probabilistic machine learning in protein ﬁtness landscape predictions"
_ICML.cc/2024/Workshop/ML4LMS — ML4LMS Poster_

### Official Review · Reviewer_CK7i · 2024-06-04
**Comprehensive evaluation of different modelling approaches**

**Rating:** 7
**Confidence:** 3

**Review:**

This is a good paper. The authors use a series of models to not only benchmark their relative performance on synthetic and real datasets, but also estimate different sources of uncertainty arising from the model and the data. I only have fairly minor suggestions for this paper

1. Given the formatting requirements, may I suggest moving at least one of figures 2 or 3 to the Appendix, and Figure 6 as well? Figure 4 instead could be worth an expansion into the two-column, larger format as I believe it makes a very nice distinction of what aleatoric and epistemic uncertainties are measuring.

2. I'm not sure if using a bog-standard MLP is a good baseline? I would have guessed a better baseline is a GP with a linear kernel than the RBF kernel (i.e. a Bayesian linear regression). Another baseline would just be a random prediction where you have an MLP that doesn't have any training at all, but makes an uneducated guess.

3. I wasn't clear on why position in the AAV dataset had anything to do with aleatoric uncertainty. Maybe this is my lack of familiarity with the dataset, but this seems like a very odd thing to test on. By extension, maybe it's worth probing whether certain types of mutations lead to greater noise (e.g. if they're destabilising or detrimental, does it lead to greater noise)?

---

### Official Review · Reviewer_uGQv · 2024-06-12
**Interesting Paper on inference of protein fitness landscapes but incomplete analysis**

**Rating:** 5
**Confidence:** 3

**Review:**

This work benchmarks six different probabilistic approaches to model protein fitness landscapes, evaluating their performance on synthetic gaussian data and experimental DMS protein datasets and using different metrics.

**Relevance**: Given the importance of learning protein fitness landscapes for protein evolution/engineering, this paper investigate an interesting issue that could contribute guidance to select the best model.

**Clarity**: The introduction and the discussion highlight well the setting of the work and the main conclusive points, but I find the paper slightly hard to follow and its readability could be improved to help the reader (eg. a table with the metrics and their definition could be useful, increasing the font size in the figures, it would be also nice to have some tables on real data in the main).

In general the work is interesting and line of research to set a standardized framework to evaluate model performances and uncertainty should be explored, yet I feel the paper is still not to a mature stage (as the authors fairly say that the analysis requires further investigation, eg. line 232-235). I understand the problem can be very much task-dependent, but the work does not offer yet any suggestion to select a model depending on the task.
The future directions discussed in the conclusion section are interesting and worth exploring, especially the role of using embeddings
to represent data when inferring a fitness landscape. It would be nice also to understand how the other methods deal with epistemic uncertainty in the OOD (as done in Fig6b for the GP): maybe the author could see when the robustness of each model breaks down depending on the distance from the ID.

---

### Official Review · Reviewer_DgVK · 2024-06-12
**-**

**Rating:** 8
**Confidence:** 4

**Review:**

-